# Could Mycolactone Inspire New Potent Analgesics? Perspectives and Pitfalls

**DOI:** 10.3390/toxins11090516

**Published:** 2019-09-04

**Authors:** Marie-Line Reynaert, Denis Dupoiron, Edouard Yeramian, Laurent Marsollier, Priscille Brodin

**Affiliations:** 1France Univ. Lille, CNRS, Inserm, CHU Lille, Institut Pasteur de Lille, U1019-UMR8204-CIIL-Center for Infection and Immunity of Lille, F-59000 Lille, France; 2Institut de Cancérologie de l’Ouest Paul Papin, 15 rue André Boquel-49055 Angers, France; 3Unité de Microbiologie Structurale, Institut Pasteur, CNRS, Univ. Paris, F-75015 Paris, France; 4Equipe ATIP AVENIR, CRCINA, INSERM, Univ. Nantes, Univ. Angers, 4 rue Larrey, F-49933 Angers, France

**Keywords:** mycolactone, analgesia, neurons, AT2R, drug development

## Abstract

Pain currently represents the most common symptom for which medical attention is sought by patients. The available treatments have limited effectiveness and significant side-effects. In addition, most often, the duration of analgesia is short. Today, the handling of pain remains a major challenge. One promising alternative for the discovery of novel potent analgesics is to take inspiration from Mother Nature; in this context, the detailed investigation of the intriguing analgesia implemented in Buruli ulcer, an infectious disease caused by the bacterium *Mycobacterium ulcerans* and characterized by painless ulcerative lesions, seems particularly promising. More precisely, in this disease, the painless skin ulcers are caused by mycolactone, a polyketide lactone exotoxin. In fact, mycolactone exerts a wide range of effects on the host, besides being responsible for analgesia, as it has been shown notably to modulate the immune response or to provoke apoptosis. Several cellular mechanisms and different targets have been proposed to account for the analgesic effect of the toxin, such as nerve degeneration, the inhibition of inflammatory mediators and the activation of angiotensin II receptor 2. In this review, we discuss the current knowledge in the field, highlighting possible controversies. We first discuss the different pain-mimicking experimental models that were used to study the effect of mycolactone. We then detail the different variants of mycolactone that were used in such models. Overall, based on the results and the discussions, we conclude that the development of mycolactone-derived molecules can represent very promising perspectives for new analgesic drugs, which could be effective for specific pain indications.

## 1. Introduction

Chronic pain is considered to be one of the most burdensome diseases in industrialized countries. Estimates of the prevalence of chronic pain range from 8% to 30% [1] and amounted to 19% in the most recent large European survey [2]. Moreover, chronic pain-related costs, reflecting medical expenses and lost productivity and income, have been estimated to more than $100 billion annually in the United States alone [3]. A total of 41% of non-cancer patients report that their pain is not under control [4], and more than 50% of cancer patients are undertreated [5]. Finally, up to 79% of patients experiencing chronic pain are unsatisfied with the management of their suffering [6,7]. Available analgesics display limited effectiveness, and, even more importantly, most effective analgesics, such as opioids, have significant side-effects and may lead to severe addiction [8,9]. Currently, a dramatic rise in the prescription of opioids have led to what is commonly referred to as the opioid epidemic, with more than 30,000 cases of death by overdose in the USA reported in 2015 [10]. Thus, the appropriate handling of pain, notably avoiding side effects and addiction as much as possible, represents an important challenge today.

The pain sensation is generated when highly specialized primary afferent nociceptors that reside in peripheral tissues transduce noxious stimuli into electrochemical impulses. Historically, nociceptors were described as the main effectors of the transduction of intense noxious stimuli [11,12,13]. As is the case for other primary somatosensory neurons, nociceptors display pseudo-unipolar characteristics, with the bifurcation of a single process from the cell body originating in the dorsal root (DRG) or trigeminal ganglion (TG) into a peripheral axon (innervating the skin) and a central axon which establish synapses on second-order neurons [14].

Interestingly, non-neuronal cells such as monocytes and macrophages also play an active role in pain through the release of inflammatory mediators [15]. Cytokines and chemokines are thus involved in mediating inflammatory responses during injury and have also been reported to mediate hyperalgesia [16]. The promotion of hyperalgesia by the interleukin IL-1β has thus long been documented [17]. Notably, it has been shown that the induction of tumor necrosis factor (TNF) alpha in inflammatory lesions contributes to inflammatory sensory hypersensitivity by inducing IL-1β [18]. With this respect, the wide use of anti-inflammatory drugs to alleviate pain further demonstrates the importance of inflammation mediators in pain [19]. Non-steroid anti-inflammatory drugs (NSAIDs), such as Aspirin or Ibuprofen, are thus widely used. These drugs are known to block the production of prostaglandins by inhibiting COX1/COX2 cyclooxygenases [20]. Furthermore, a cross-talk between the nociceptive and immune systems has been reported to elicit pain, with the subject receiving growing attention [21,22]. Thus, nociceptors express receptors that are activated by inflammatory cytokines [21,22]. Nociceptors release the neuropeptide substance P, along with calcitonin-gene related peptide (CGRP), modulating both the recruitment of immune cells and their production of inflammatory mediators [21,22].

Two principal categories of sensory receptors, ion channels and G-protein-coupled receptors (GPCRs), detect thermal, mechanical or chemical stimuli and mediate nociception [11,23] (Figure 1). Ion channels are activated by numerous noxious stimuli from many sources [24]. Therefore, ion channels play a pivotal role in the mediation of pain [12] and represent relevant targets for the development of new analgesic drugs [25]. Indeed, drugs targeting ion channels are less likely to lead to addiction than currently used drugs targeting GPCRs [26,27]. For example, the TRPV1 receptor (transient receptor potential cation channel vanilloid subfamily member 1), a Ca2+ permeable, non-selective cation channel that is expressed in the plasma membrane of sensory neurons, is activated by various noxious signals (including high temperature, low pH, chemicals or inflammatory mediators), mediating the sensation of pain via calcium influx [28]. Interestingly, IL-1β activates interleukin 1 receptors (IL-1R1) on nociceptor neurons, thus increasing TRPV1 expression and, accordingly, pain sensitivity to thermal stimuli [22].

Notably, capsaicin, the main active ingredient of hot chili peppers, is a potent, highly selective TRPV1 agonist, and a capsaicin 8% patch relieves pain in patients with peripheral neuropathic pain [29] (Table 1). Interestingly, NSAIDs also appear able to desensitize TRPV1 channels—a mechanism supposed to participate in the NSAID-mediated attenuation of hyperalgesia [30].

Opioids, which are widely used analgesic drugs, bind to receptors belonging to the GPCR family and are very efficient in alleviating pain, but the activation of some opioid receptors also mediates numerous adverse effects such as respiratory depression, sedation, constipation, nausea, vomiting, reward/euphoria and dependence/withdrawal symptoms, thus restricting the use of opioids in pain therapy [31]. Aside from the opioid receptor system, increasing interest is given to angiotensin (Ang) II type 2 receptor AT2R, another GPCR that was shown to be directly involved in pain pathways. Although still debated, AT2R was shown to be expressed in human sensory neurons and, upon activation, to mediate thermal pain via the sensitization of TRPV1 ion channels [32,33]. AngII, the endogenous ligand for AT2R, acts in vitro on the receptor, leading to the sensitization of TRPV1 and increased neurite outgrowth in DRG neurons. In this context, EMA401, a high-affinity ligand for AT2R [32], was shown to counteract AngII signalling and was thus considered a valuable candidate for pain management (reviewed by Smith and Muralidharan [34]) (Table 1).

Current analgesics fail in terms of their efficacy and safety, and accordingly, alternative strategies are in high demand [35]. In this context, nature-inspired pain-killing solutions are gaining increasing interest [36]. Fortunately, the exploitation of extracts of Chinese medicinal ants that display analgesic and anti-inflammatory properties has begun [37]. Interestingly, plant, animal or bacterial toxins affect ion channels with high potency and selectivity [38,39]. Thus, they represent a powerful tool for the identification of novel molecular targets and potentially for the design of novel agents for pain relief [38]. For example, the neurotoxic protein *Botulinium* toxin type A (also called the “miracle toxin”), produced by the bacteria *Clostridium botulinum*, was shown to alleviate pain in cervical dystonia [40] and migraine [41] (Table 1). Likewise, cone snails, also known as marine gastropods, produce in their venom small, pharmacologically active peptides, referred to as conotoxins. These toxins exhibit a strong inhibitory effect on ion channels, transporters and receptors (reviewed by Lewis [42]), showing a potent anti-nociceptive effect in pre-clinical models of chemical and injury-induced pain [43,44]. 

Against this general background, the attempt to take advantage of a surprising characteristic feature of Buruli ulcer (BU), a neglected tropical skin infectious disease caused by *Mycobacterium ulcerans* [45]—namely the painlessness of the lesions—seems particularly promising. Indeed, mycolactone, the toxin produced by *M. ulcerans*, is responsible for the extensive ulcerative skin lesions observed in patients suffering from BU [46,47,48] but also for the painless character of the lesions it causes, at least at early stages of the disease [49,50]. In mice models, mycolactone induces hypoesthesia with an extremely long-lasting effect [51,52,53]. Using chemical synthesis approaches of mycolactone, but also various cellular and in vivo models, several molecular targets have been proposed for mycolactone that would account for its analgesic properties (Figure 1), but their relative contributions to the different effects of mycolactone remain controversial. The detailed elucidation of the molecular pathways underlying the various effects of the mycolactone toxin could then pave the way for the design of novel potent analgesic compounds.

## 2. Mycolactone-Induced Analgesia: Inflammatory Versus Non-Inflammatory Contexts

The pathogenesis of BU is dependent on the toxin mycolactone secreted by *M. ulcerans*. Mycolactone exerts pleiotropic effects and disrupts fundamental cellular processes, such as cell adhesion, signaling pathways ([54]; reviewed by Sarfo and colleagues [55]) or response to stress [56], playing a central role in host colonization [46,57]. More importantly for this review, mycolactone is also responsible for the hypoesthesia in BU [58]. Several cellular targets were proposed to account for the various effects of mycolactone. The first was the Wiskott–Aldrich Syndrome protein (WASP) reported by the Demangel group [59,60]. Indeed, this group showed that mycolactone binds to WASP, with an affinity 100-fold greater than its natural activator, leading to alterations in actin dynamics, defective cell adhesion and eventually apoptosis [60]. The mycolactone-driven destruction of cutaneous tissues was thus suggested to be caused by hyper-activation of neural (N)-WASP. The second target that was identified by Simmonds et al. is the core subunit of the Sec61 complex, which is now recognized as the primary cellular target of mycolactone [54,55,56,59]. As Sec61 is responsible for the transport of newly synthesized proteins through the endoplasmic reticulum membrane, the toxin was shown to impact the expression of a set of proteins such as immune mediators [61,62,63]. In parallel, the groups of Altmann and Pluschke showed that mycolactone acts as an inhibitor of the signalling of the target of rapamycin (mTOR), preventing Bim-dependent apoptosis [64]. For instance, mycolactone was shown to inhibit the mTOR-dependent phosphorylation of the serine/threonine protein kinase Akt, leading to the up-regulation of transcription factor Fox03 and the increase of Bim expression. The increase of Bim expression leads to cell apoptosis. This signaling was shown to be important in BU pathogenesis, with Bim-knockout mice not developing necrotic lesions of BU when infected with *M. ulcerans* [64]. Simultaneously, the groups of Brodin and Marsollier identified AT2R as the main receptor involved in mycolactone-mediated analgesia [52]. More recently, a modelization study predicted that mycolactone would bind to Munc18b, a key chaperone protein of blood platelet degranulation, blocking exocytosis and hence impairing the release of compounds and factors that are critical for wound healing processes [65]. In the present review, we describe the mechanisms proposed for sustaining mycolactone-induced analgesia.

Experimental infection with *M. ulcerans* in a mouse footpad model using the response to mechanical noxious stimulus in the Von Frey filaments test and foot retraction demonstrated the painlessness of the lesions [66]. In addition, a reduced perception of pain was observed in mice, with mycolactone injected directly into the footpad, with a delayed response to the same noxious mechanical stimulus [51] (Table 2). These previous works on mycolactone suggested that mycolactone-induced hypoesthesia was attributable to nerve destruction at late stages of the disease [51,66]. It was then proposed that the underlying mechanism was relevant to cytotoxity, as in leprosy [45]. However, such a mechanism could not account for the analgesic effects observed at earlier stages, before the occurrence of tissue destruction. In this context, it was then natural to ask whether mycolactone could interfere with neural transmission [52]. In contrast with the results of previous work [51], it was shown in Marion et al. (2014) [52] that mycolactone activates AT2R in neurons, with the release of phospholipase A2-mediated arachidonic acid (AA), the generation of prostaglandin E2 from AA by cyclooxygenase-1 and the subsequent release of potassium through TRAAK channels. Notably, drugs known to activate potassium channels display antinociceptive properties [67] as they induce hyperpolarization, thus decreasing neuron sensitization. The sustained hyperpolarization induced in sensory neurons was proposed to mediate the analgesic properties of mycolactone, in response to a noxious thermal stimulus ([52] (tail-flick adapted test), [53] (Hargreaves plantar test), [68]). In support of this result, hypoesthesia was reported to be dependent on the expression of AT2R, as the genetic knockout of this gene resulted in the restoration of a normal latency period for the sensing of thermal pain in mice [52]. The return to normal sensitivity after this period also proved the absence of nerve destruction in this mouse model.

To further assess the relevance of this suggested novel signalling pathway, the investigation of the effect of mycolactone—originally conducted in macrophages, PC12 pheochromocytoma cells, and hippocampal neurons [52]—was further extended to dorsal root ganglion neurons (DRG neurons). The study demonstrated that in DRGs, mycolactone also triggers hyperpolarization through AT2R [69]. Interestingly, mycolactone shares a common characteristic with the neuroprotective agent riluzole that displays anesthetic properties at high concentrations [70] and was shown to activate TRAAK potassium channels [71]. Together with the TREK1 potassium channel, TRAAK was shown to be a critical regulator of nociceptor activation [72]. Accordingly, the functional activation of potassium channels appears to be efficient in alleviating pain [73], and the hyperpolarization of neurons via the activation of potassium channels appears to represent a common characteristic for both opioid and non-opioid analgesic drugs [74,75].

The activation of AT2R by mycolactone was thus shown to mediate its analgesic properties [52,69]. AT2R is an unusual GPCR, with a lack of observed G proteins or the recruitment of β-arrestins and finally the absence of intracellular internalization. Accordingly, the first report of an AT2R structure bound to a ligand showed that helix VIII was found in a non-canonical position stabilizing the active-like state and thus preventing coupling to G-proteins [76]. It is thus difficult at present to undertake mechanistic studies to compare the pharmacology of the different known ligands of AT2R. Danser and Anand showed that the AT2R signalling cascade in sensory neurons elicited peripheral pain sensitization and that the AT2R ligand EMA401 alleviated pain in preclinical models [77,78]. Pain relief was further demonstrated in a phase II clinical trial involving patients with neuropathic pain associated with postherpetic neuralgia [33,79]. As a difference to EMA401 or other known ligands of AT2R (including the endogenous peptide AngII or the C21 agonist), only mycolactone appeared able to trigger the hyperpolarization of DRG neurons [32,69,80,81]. Moreover, EMA401 or C21 did not inhibit hyperpolarization triggered by mycolactone [69] (Table 1). This observation pinpoints the specificity of mycolactone/AT2R interactions, supporting the hypothesis that the AT2R binding site of mycolactone is different from that of AngII, C21 or EMA401.

In a parallel study aimed at deciphering the mechanisms responsible for mycolactone-induced hypoalgesia, Anand and colleagues [80] examined the effect of mycolactone treatment on calcium influx via the TRPV1 receptor after the stimulation of DRG neuronal cultures with the noxious agent capsaicin. As mentioned in the introduction, TRPV1 is involved in the integration of diverse painful stimuli [82]. It was recently reported that the capsaicin-evoked action would follow a physical interaction between TRPV1 and anoctamin 1, a calcium-activated chloride channel, following the entry of Ca2+ through the TRPV1 pore, with such an interaction being relevant for the enhancement of nociception [83] (Table 1). Capsaicin responses in human and rat DRG neurons were dose-dependently inhibited in the presence of mycolactone [80], suggesting that channels other than potassium channels could be targeted by mycolactone. Capsaicin is used in modelling nociception through TRPV1 activation, but also demonstrated analgesic properties when used at low concentrations in topical formulations [84], demonstrating the singularity of ion channels in pain management. Also, in agreement with the conclusions of Song et al. (2017) [66], the study by Anand et al. (2016) [80] showed that the morphological and functional effects of mycolactone were not affected by angiotensin II or AT2R antagonist EMA401. These observations again demonstrate the specificity of the interactions involved in the mycolactone–AT2R pathway, also highlighting the need for further investigations to disentangle the underlying mechanisms. Such investigations appear all the more important as there is now contrasting evidence regarding the expression of AT2R in DRG neurons. Indeed, in contradiction to results of Anand et al. (2013, 2015, 2016), Song et al. (2017) or Benitez et al. (2017) [32,69,78,80,85], Shepherd and co-workers (2018) [86] have found that AT2Rs were not expressed in DRG neurons; rather, this study showed that the action of angiotensin II on macrophages leads to the activation of mouse and human DRG sensory neurons [86]. Indeed, in the case of tissue injury, the activation of AT2R in macrophages would trigger the production of reactive oxygen species, which would lead to sensory neuron excitation. Therefore, a proposed mechanism for the analgesic effects exerted by AT2R ligands, such as EMA401, was the blockade of the production of macrophage-derived ROS/RNS, resulting in a decreased pathological excitation of mouse and human sensory neurons [86]. Such findings provide radically different insights into the understanding of the role of angiotensin signaling in pain sensitization. It then appears particularly important at this stage to characterize the interaction between mycolactone and AT2R. To this end, an extensive search for AT2R ligands in various collections of chemical libraries should be performed to investigate their ability to induce hyperpolarization—an effect that could be then further assessed on primary murine DRG sensory neurons [73].

A typical feature of mycolactone is immunity modulation [87], and the intrathecal injection of mycolactone in rats down-regulated the basal production of inflammatory cytokines in the spinal cord [63]. By investigating the mechanisms involved in the inhibition of cytokine production by mycolactone, the toxin was shown to block the translocation of secretory proteins into the endoplasmic reticulum (ER) in host cells. Lines of evidence were provided in support of the ability of mycolactone to prevent the translocation of many cytokines across the ER, inducing their subsequent degradation in the cytosol [88,89,90] and leading to defective inflammatory responses [86]. Sec61 translocon, a heterotrimeric membrane protein complex essential for protein translocation into the ER, was thus identified as a major target of mycolactone [88], and its inhibition by the toxin was shown to have critical consequences on the immune system [61]. It has also been proposed that the blockade of Sec61 by mycolactone could contribute to mycolactone-induced analgesia by suppressing inflammation, and further investigations are definitely needed [63]. The anti-nociceptive activity of mycolactone was also demonstrated in the formalin assay in mice, which recapitulates an inflammatory context [62]. The injection of formalin solution into mouse hindpaws induces two distinct periods of stereotypical licking and biting behaviors and allows discrimination between inflammatory and non-inflammatory pains. Intraperitoneal injection of mycolactone 1 h before formalin injection had no effect on the early phase (5 min), reflecting a direct effect on nociceptors (acute pain), whereas in the second phase, lasting from 10 to 40 min, an anti-nociceptive effect was observed, involving primarily inflammatory processes [62] (Table 2). Guénin-Macé and colleagues suggested that the inhibition of Sec61 activity by mycolactone could account for its hypoesthesic effect. Indeed, the interaction between mycolactone and Sec61 was shown to promote cell death via a reticulum endoplasmic stress response triggering cell apoptosis [56] and also to lead to an impaired production of key mediators of immune responses [61,63].

The effect of mycolactone on the initiation of the stress response is rapid, at within 1 h, comparable to the hypoesthesic effect of mycolactone, which is detected 2 h after mycolactone injection, arguing in favour of the involvement of such a broad-spanning pathway in the hypoesthesia [56]. However, it is important to emphasize that mycolactone-mediated Sec61 blockade was studied in an inflammatory context and often after a prolonged exposure of cells with the toxin [63,91], or in studies of pathways related to cellular stress responses [56]. Furthermore, the blockade of Sec61 may alter the functional biology of sensory neurons beyond inflammation and interfere with processes of protein expression and signalling [91]. Nevertheless, it appears that the expression of AT2R is not affected by mycolactone-mediated Sec61 blockade [91], suggesting that the effect of mycolactone on AT2R is independent of Sec61. Further experiments, with an in vivo model of acute pain, will thus be needed to assess whether Sec61 exerts a role in mycolactone-induced analgesia.

## 3. Detection and Synthesis of Mycolactone: Key Starting Points for the Development of Mycolactone-Inspired Potent Analgesics

The existence of a virulence factor or toxin contributing to the extensive ulcerative lesions caused by *M. ulcerans* has been suggested since 1965 [92]. After several unsuccessful attempts to isolate the toxin, it was first purified in 1998, and its lipid nature was determined by Small and colleagues [93].

Lipid extraction from *M. ulcerans* extracts, according to the methanol/chloroform/water Folch’s extraction method, allows the separation of the aqueous phase from the organic one containing the lipids. Purification then proceeds by liquid chromatography coupled to mass spectrometry analysis [46,94,95,96,97]. The purified toxin was identified as a 12-membered ring macrolide core with two polyketide-derive side chains and called mycolactone (to reflect its mycobacterial source and chemical structure). It was then possible to demonstrate the role of the lactone exotoxin in the pathogenesis of BU [46].

The structure of mycolactone A/B has thus been determined as a 3:2 equilibrating mixture of mycolactones A and B major and minor Z-Δ4′,5′- and E-Δ4′,5′-isomers (Figure 2), respectively, in the unsaturated fatty acid side chain [94,98]. Mycolactones A and B generated an overlapping two-peak cluster and were, therefore, quantified together. MS analyses of the mycolactone A/B standard led to the formula C44H70O9 for the compound and revealed the presence of an ion with a mass-to-charge ratio (m/z) of 765.5, corresponding to sodium adduct [M+Na]^+^, as the main component. The selected reaction monitoring (SRM) of this ion yielded an ion with an m/z of 429.2, corresponding to the core lactone ring of mycolactones [94]. 

Interestingly, mycolactones share chemical structure similarities with other lactone-containing natural products such as sesquiterpene lactones that also display analgesic properties [99]. In line with this, plants of the Asteraceae family have been used in traditional medicine for centuries for their analgesic effects, and extracts from the *Arnica montana* plant displayed very interesting analgesic properties [100]. Notably, the consumption of Kava plants showed beneficial effects including analgesia, with such an effect being attributed to lactones [101]. It is also notable that Tilmicosin, a macrolide antibiotic, shares structural properties with mycolactone [102]. This compound was demonstrated to display anti-inflammatory potential, and its possible pain-relieving effect was investigated [102]. Contrary to mycolactone, tilmicosin was capable of attenuating chemical-induced, but not thermal-induced, acute pain in mice [102]. The structure of the mycolactone toxin could thus account for its analgesic properties. The production and purification of mycolactone from *M. ulcerans* extracts (Figure 2) require large amounts of bacteria, as only a few micrograms of mycolactone are obtained from 100 mg of bacteria [94]. Also, the culture of *M. ulcerans* is a fastidious process: primary cultures are usually positive within a 6–12-week incubation, with the obtention of isolates, however, sometimes requiring much longer periods of time of up to nine months [103]. A recurrent idea since the discovery of the giant plasmid pMUM001 by Stinear and Brosch, harbored by *M. ulcerans,* which encodes mycolactone [104], was to express the plasmid in the faster-growing and closely-related mycobacterium *Mycobacterium marinum*. However, the genetic background of *M. ulcerans* appeared to be important for the production of mycolactone as, despite the efficient production of mycolactone polyketide synthases (PKS), whole mycolactone could not be detected as properly expressed in *M. marinum* [105]. In line with this, the same group led by Stinear reported that fast-growing *Mycobacterium smegmatis*, which is a widely used host for mycobacterial product expression, was unsuitable for the efficient replication of the PKS-encoding plasmids required for mycolactone synthesis [105]. To our knowledge, no successful expression of mycolactone in *Escherichia coli* has been reported.

Also, mycolactone is only soluble in organic solvents due to its lipid nature, with appropriate storage and handling conditions being important for its proper conservation [95,106]. Nonetheless, despite these limitations, the bio-extraction of mycolactone can be considered overall to be an easy process, and notably, high solubility was demonstrated for mycolactone in corn oil with 8% ethanol [53] (Figure 2).

Heterogeneity in the structure of mycolactones has been demonstrated, depending on *M. ulcerans* strains, with possible implications for virulence [94,107,108] (see also [109] for review). The total biosynthesis of mycolactone allowed the confirmation of its structure [110] (Figure 2). Mycolactone-like metabolites were also identified during the biosynthesis of mycolactone [94] and isolated from *Mycobacterium* ssp [109]. Also, mycolactone-like metabolites are produced by other mycobacteria fish and frog pathogens, referred to as “mycolactone-producing mycobacteria”, such as *M. liflandii*, *M. pseudoshottsii*, or *M. shinshuense* [111,112,113]. These strains, considered to be *M. ulcerans* ecovars, could be considered for mycolactone production, but mycolactones produced by these strains appear to be structural variants of the mycolactone A/B produced by *M. ulcerans* [114]. *M. liflandii*, *M. pseudoshottsii*, or *M. shinshuense* thus produce mycolactone E, F, S1 and S2 [111,112,113], and it would be necessary to further assess these different mycolactones for their analgesic properties. Notably, the production of active metabolites displaying analgesic properties could account for the maximal analgesic effect obtained 6 h after the injection of purified mycolactone, while very low (+/− 1%) amounts of mycolactone were recovered in injected footpads [53]. In this respect, it is also noticeable that metabolites of various painkillers were reported to display potent analgesic properties [115]. Novel synthesis approaches were then used to design various mycolactone analogs [94,116,117], as reviewed by Gehringer and Altmann (2017) [118], and also to elucidate the structures involved in the triggering of the biological effects of mycolactone in structure–activity studies [119]. Interestingly, mycolactone structural variants displayed significantly different immunosuppressive activities [120], with the loss of hydroxyl groups reducing the cytotoxicity of mycolactone [119]. Thus, mycolactone C, a natural derivative of mycolactone A/B lacking a hydroxyl group, as well as the synthetic structural variant devoid of all the hydroxyl groups in the lower side chain PG155, displayed a reduced toxicity along with a reduced immunosuppressive effect [119]. In comparison to mycolactone A/B, no changes in cell morphology, proliferation or metabolism were observed after treatments with derivative compounds with significantly truncated lower side chains [119], demonstrating the potential of these compounds in strategies aimed at alleviating pain. The synthesis of mycolactone is a complex process, requiring at least 20 steps for the biosynthetic engineering of the core alone [118] (Figure 2).

## 4. Perspectives for the Use of Mycolactone as an Analgesic

Pain is the hallmark of many infections. However, in contrast to typical painful microbial infections, some pathogens take advantage of the analgesia they implement to facilitate their spread. Such is the case for *M. ulcerans*. *M. leprae*, the causative agent of leprosy, similarly causes hypoesthesia, which is presumably secondary to the extensive nerve damages it induces [21]. Here, it appears important to highlight the fact that in recent reviews presenting the molecular mechanisms of microbial-driven pain, mycolactone was the only bacterial toxin that silences pain through an inhibition of pain signaling [15,21].

As pointed out above, various mechanisms of action have been proposed to account for the analgesic properties of mycolactone. As compared to some other natural models of analgesia (concerning, for example, scorpions [121]) the triggering of the relevant signal by mycolactone involves a full cellular pathway, instead of a direct action on the ion channel involved in the signaling. Indeed, it was shown that mycolactone triggers neuronal hyperpolarization by acting on AT2R receptors [52,69]. Thus, the hyperpolarization, with the opening of TRAAK potassium channels, involves a full cellular pathway, which does not appear to be targeted by other currently known analgesics. It is then all the more important to further dissect, in as much detail as possible, the underlying cellular pathway, namely phospholipase A2 (PLA2), cyclooxygenases and prostaglandins (PGE2). It was also shown that mycolactone reduces inflammatory mediators through the blockade of Sec61 [61,63]. Is it still unclear whether these two systems are differently activated, depending on the noxious stimulus and the inflammatory context, or could be rather involved in the cross-talks important for mycolactone-induced analgesia (see Guénin-Macé et al., 2019 [122]). Such a possibility is worth considering, as it is now well acknowledged that pain and inflammation are related conditions [21,22]. Thus, via its action on immunity, mycolactone could block the cross-talk between the immune system and nociceptors, with a resultant decrease in the excitability of neurons. Notably, the pathways triggered by mycolactone are not those involved in the side effects of common analgesics [31].

From the perspective of using mycolactone (or derivatives) as an analgesic, it is important to stress that induced toxicity was observed only for high concentrations (>5 µg) or high incubation times (>24 h) ([69], Table 1). Furthermore, remarkably, 80% of DRG neurons remained viable after 48 h of incubation with 70 µM mycolactone [69]. The fact that the hypoesthetic effect of mycolactone is of limited duration, disappearing after 48 h, gives further weight to the absence of toxicity at the doses used [53]. For the production of mycolactone-inspired pain-killers, the two possible ways of obtaining mycolactone—namely bio-extraction and synthesis—offer interesting perspectives. In this respect, various kinds of excipients could be tested, in line with a recent work [106] aimed at optimizing the bioavailability and efficacy of purified mycolactones. A striking characteristic of *M. ulcerans* is its ability to assemble into a biofilm, as first observed on the surface of aquatic plants [123]. For many human bacterial pathogens biofilms consist of discrete bacteria surrounded by an extracellular matrix (ECM), whereas the ECM of *M. ulcerans* is associated with only the outermost cell layer [124]. Interestingly, the ECM of *M. ulcerans* appears to contain vesicles which were identified as a reservoir of the mycobacterial toxin [124]. The encapsulation of extracted and purified mycolactone into polymers which are able to improve the bioavailability of poorly soluble drugs could thus represent an exciting possibility for obtaining stronger and more prolonged analgesic effects in vivo [125].

Also, in line with the investigations into deciphering the respective roles of the core and the side chains of mycolactone in the various biological effects of the toxin [119,120], the study by Guénin-Macé and co-workers [62] tried to assess whether mycolactone could be engineered to allow the dissociation of immunosuppression and analgesia from cytotoxicity. As a matter of fact, such strategies have previously been successfully developed and implemented for other toxins. Thus, slight modifications of the structures of conotoxins make them devoid of adverse effects [126]. Thus, it was shown that the synthetized conotoxin Xen2174 increases the action of noradrenaline, inhibiting the sensation of pain. This compound has been already successfully tested to alleviate chronic pain in cancer patients and, unlike opioids, did not cause addiction. This class of peptides is easy to synthesize and can be administered orally, provided their structure is slightly modified to avoid degradation by digestive enzymes. Some conotoxin derivatives are used in the clinic as novel pain killers under the name Prialt (ziconotide) [127] (Table 1). Mycolactone-like molecules were therefore investigated as drug candidates against chronic skin inflammation ([62], reviewed by Sarfo and colleagues in 2016 [55]). Notably, a variant of mycolactone (5b; devoid of north chain and core C8-methyl), displayed optimal characteristics in terms of immune suppression, without cytotoxicity. Remarkably, this compound displayed enhanced binding to AT2R and inhibited inflammatory cytokines responses [62].

Recent studies provide a proof-of-concept that mycolactone-derived compounds have a therapeutic potential in treating pain [62,69]. In addition, biosynthesis studies have demonstrated that mycolactone-like compounds with a truncated/modified structure lost cytotoxicity compared to whole mycolactone [117,119,120]. It would then be interesting to test the potential analgesic effects of these compounds in pain models.

In addition to the safety of topical administrations, a local administration of mycolactone appears the most suited for mimicking its natural mode of action, notably because of its rapid elimination and low diffusion from the site of inoculation [53]. For instance, in the case of severe burns, skin is removed from one area of the body (donor) to be transplanted to the burned region. Pain at the skin donor site is reported to be one of the most distressing symptoms during the early postoperative period. Besides, improper tissue adhesion provokes irritations and tensions, which are another source of pain. In such situations, analgesics such as lidocaine and bupivacaine are administered to relieve pain, but their action lasts only for a few hours [47], and the doses needed are very high (25 mg of lidocaine per local application) [128]. Capsaicin cream is increasingly used, significantly improving pain relief, including surgical pain; however, it has significant side effects [129]. Many side effects are also reported for non-steroid anti-inflammatory drugs such as diclofenac (Voltaren), such as cardiovascular or hemorrhagic risks [130]. Following liver resection, pain can last for several weeks [131]. In this case, and also for other surgeries, post-operatory pain can be very intensive. As reported in several studies, the pain is highest during the first 24 h after the surgical procedure [132]. Against such a background, several properties associated with mycolactone appear of particular interest, with a long-lasting analgesic effect already observed after only 2 h. Optimal postoperative pain control is particularly needed for early mobilization and improved respiratory function, implying the use of opioids. The most common opioids used are morphine, hydromorphone and fentanyl, but they display severe side effects. In addition, for comparable analgesic effects, the required doses of morphine are 300 times higher than those of mycolactone [133]. Moreover, cirrhotic patients display an increased bioavailability of opioids and benzodiazepines, due to decreased drug metabolism in the liver, and as a result, drug accumulation can occur. In this context, there is intensive research in the field regarding the use of local anesthetic infusions and developing novel drugs to relieve pain [134,135]. In addition to its very rapid action, mycolactone is active at very low doses, since a decrease in the response to noxious heat stimulus was already observed in mice 4 h after an injection of 1 µg of purified mycolactone [53]. The hypoesthesia effect of mycolactone was maximal 6 h after injection of 4 µg with very low (+/− 1%) amounts of mycolactone recovered in injected footpads over the same time. Importantly, the hypoesthesia effect lasted for more than 24 h, highlighting the potential strengths of mycolactone as a potent analgesic [53]. In summary, mycolactone represents a valuable candidate for local pain management, in a context where existing topical treatments appear ineffective [136].

## 5. Conclusions

The management of pain represents a challenge, as available treatments have limited efficacy and display many side effects, such as addiction in the case of opioids. New potent therapeutic strategies are highly needed, with higher benefit/risk ratios, with the aim of long-lasting analgesia effects obtained with low doses. Inspiration from Mother Nature represents a good starting point for the discovery of novel high-efficacy analgesics, devoid of adverse effects as far as possible. Mycolactone, the lactone toxin produced by *M. ulcerans*, is responsible for the painless ulcerative skin lesions observed in patients suffering from Buruli ulcer. Here, we have described the extremely long-lasting hypoesthetic effect of mycolactone, observed in inflammatory and non-inflammatory models, at low doses. The immunomodulatory effect of mycolactone, as well as a direct action on AT2R receptors, could account for the analgesic properties of the lactone, depending on the inflammatory context. A cross-talk between these two systems could be also involved in mycolactone-induced hypoesthesia. The detailed understanding of this cross-talk could further provide powerful ways to target the overall system, with the nociceptive and immune components, with increasingly more potent compounds.

## Figures and Tables

**Figure 1 toxins-11-00516-f001:**
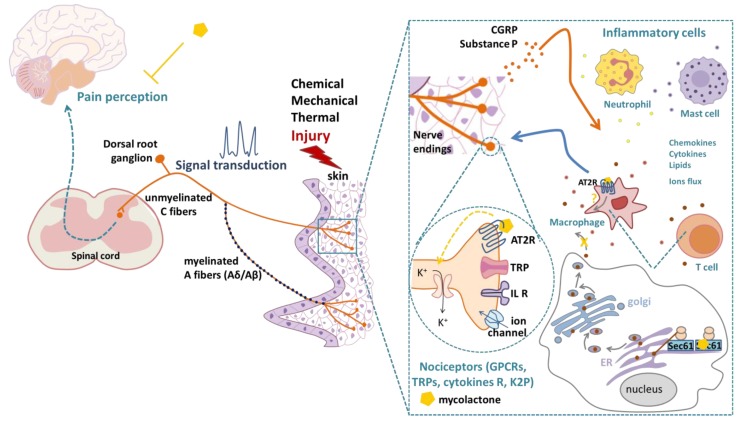
Mechanisms of pain perception upon a skin injury and reported effects of mycolactone on different cell types. Pain perception is mainly mediated by the signal transduction of nociceptors with cell bodies located in the dorsal root ganglion and axons of the Aδ- or C-fiber type extending to the skin. Inflammatory cells (neutrophils, mast cells, T cells, macrophages) release mediators (chemokines, cytokines, lipids, interleukins (IL)) that are detected by nociceptor terminals to modulate neuronal excitation and the transduction of pain signals. Immune mediators can also impact ion channel trafficking to the membrane or ion channel transcriptional expression. The overall result of these immune-mediated pathways in nociceptors is the lowering of the threshold for responses to an external stimulus, leading to increased pain sensitivity. Nociceptors release substance P and CGRP (calcitonin-gene related peptide), which act on inflammatory cells leading to the release of inflammatory mediators. Mycolactone causes neuronal hyperpolarization by inducing angiotensin II receptor 2 (AT2R) signaling. Mycolactone also induces a Sec61 translocon blockade and inhibits protein translocation into the endoplasmic reticulum (ER). The inhibition of Sec61 thus prevents the production of key mediators of innate and adaptive immune responses. AT2R and Sec61 were both shown to mediate the mycolactone-induced inhibition of pain perception. K+: potassium, GPCR: G-protein coupled receptor, TRP: transient receptor potential ion channel, R: receptors, K2P: two-pore domain potassium channel.

**Figure 2 toxins-11-00516-f002:**
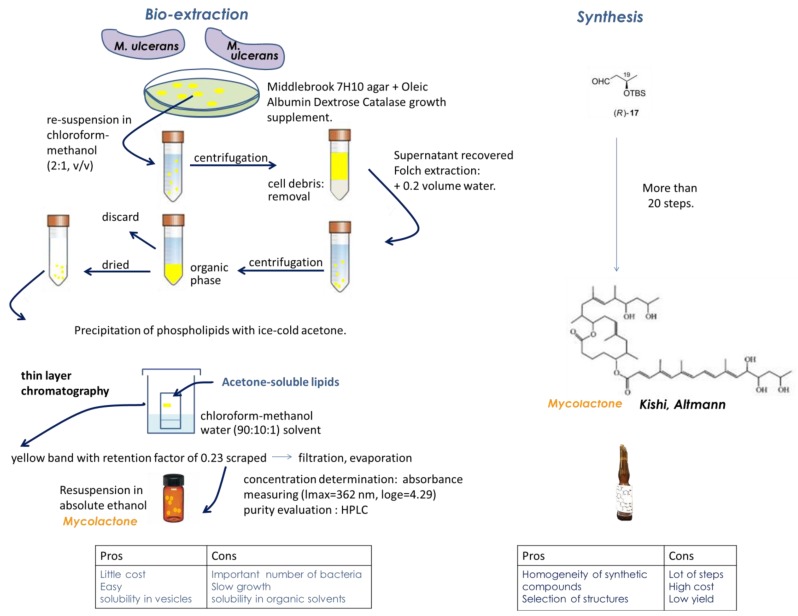
Mycolactone bio-extraction versus synthesis: pros and cons.

**Table 1 toxins-11-00516-t001:** Molecules targeting AT2R or ion channels: structures and activities.

Molecule	Structure	Biological Effect
Angiotensin II	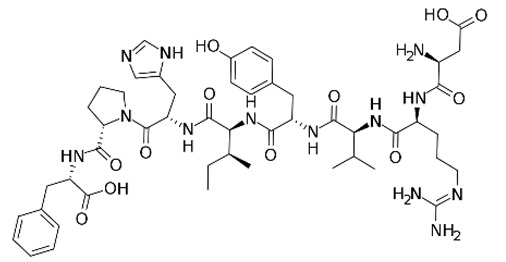	Natural AT1R and AT2R agonist, involved in haemodynamic effects. AngII induces TRPV1 sensitization, promotes pro-inflammatory responses, regulates gene expression and participates in many cell processes and signalling pathways.
Mycolactone	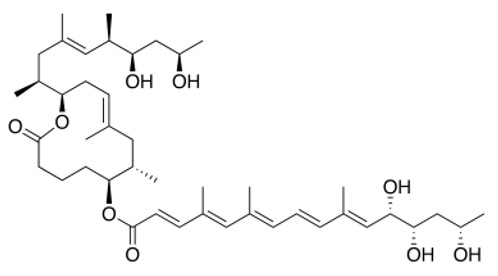	AT2R ligand. Main factor of virulence of *M.ulcerans*; the toxin induces analgesia, sustaining the painlessness of the ulcerative lesions in Buruli ulcer (BU). Analgesia is mediated by K^+^-dependent hyperpolarization through AT2R activation. Sec61-dependent anti-inflammatory activity on the immune and nervous systems could also contribute to BU-associated analgesia.
Compound 21 (C21)	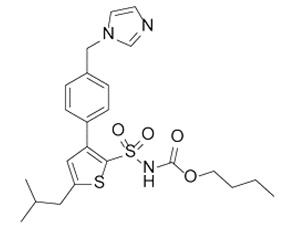	First synthetic selective agonist of AT2R; therapeutic potential for heart failure, nephroprotection, anti-inflammation, stroke and some dermatological applications. Does not interfere with mycolactone in binding to AT2R.
EMA401	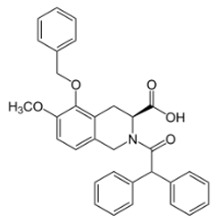	AT2R antagonist. Potent pain killer. Decreases TRPV1 expression. In a phase II clinical trial, the efficacy of EMA401 (100 mg, twice a day) was demonstrated in neuropathic pain, in comparison to placebo, after 3–4 weeks of oral administration.
Capsaicin	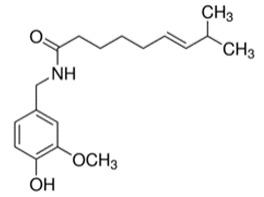	Activates TRPV1. Used in chemical pain models. At low doses, promotes analgesia through the desensitization of the TRPV1 receptor.
Conotoxin	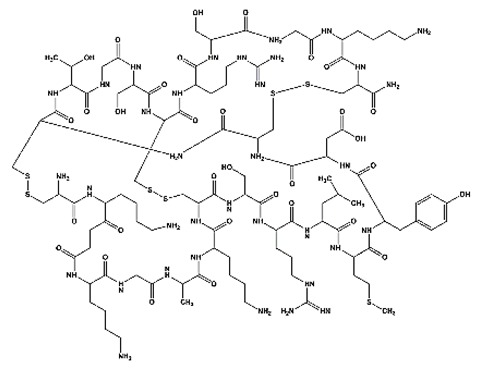	Toxin extracted from cone snails. Promotes analgesia. Inhibits the release of pro-nociceptive neurochemicals such as glutamate, CGRP, and substance P. Ziconotide (Prialt) is the synthetic compound inspired by conotoxins. This compound is an atypical agent for the management of severe and chronic pains. Intrathecal (IT) administration of Ziconotide appears 1000 times more effective than IT morphine delivery.
Botulinium toxin A	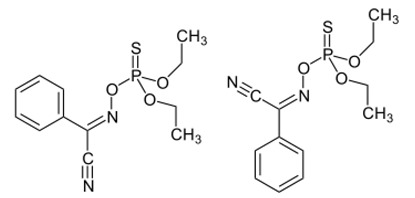	Toxin extracted from *Clostridium botulinum*, induces analgesia. Prevents the release of neurotransmitters CGRP and substance P and the expression of TRPV1.

**Table 2 toxins-11-00516-t002:** Mycolactone-induced analgesia: overview of the effects of mycolactone in non-inflammatory vs inflammatory models of pain.

Pain	Mycolactone-Induced Analgesia				
	Pain Model	Dose-Time Effect	Suggested Mechanism	MycoL	Ref.
Non inflammatory pain	Von Frey filament mechanical pain	100 µg into mouse footpad, 28 days: ↓response to pressure (g)	Nerve damage	Purified strain 1615	[51]
	Tail-flick adapted thermal pain	5 µg into mouse footpad, 2 h→48 h:↑latency in withdrawal of the footpad in response to noxious thermal stimulus	K^+^-dependent hyperpolarization of neurons through AT2R, a mechanism further confirmed in DRG neurons [69]	Purified strain 1615, ∑ mycoL	[52]
	Hargreaves plantar test thermal pain	1,2,4 µg into mouse footpad:↑latency in withdrawal of the footpad in response to noxious thermal stimulus, analgesia 2 h→48 h for the highest dose	Basal state reached after 48 h indicating no nerve damages; K^+^-dependent hyperpolarization of neurons through AT2R	Purified strain 1615	[53]
	Early phase of formalin-induced chemical pain (0–5 min)	IP injection of mycoL purified (0.5 mg/kg) or 5b (5 mg/kg) 1 h before subcutaneous injection of 10 µL of formalin solution (5%) into mouse hindpaw: no↓in pain score (paw licking duration and body tremor number)	Efficiency at inflammatory stages	Purified strain 1615, ∑ 5b	[62]
Inflammatory pain	Second phase of formalin-induced chemical pain (10–40 min)	IP injection of mycoL purified (0.5 mg/kg) or 5b (5 mg/kg) 1 h before subcutaneous injection of 10 µL of formalin solution (5%) into mouse hindpaw: significant↓in pain score anti-inflammatory effect on DRG neurons stimulated by LPS (16 h)	Analgesic effect via AT2R, suppression of inflammatory cytokine production, or both mechanisms	Purified strain 1615, ∑ 5b	[62] [63]
Neuropathic pain	Chronic constriction injury of the sciatic nerve	Intrathecal injection of 100 ng mycoL 3 days to rats at day 2 post-operation: ↓expression levels of pro-inflammatory cytokines in mycoL-treated sham-operated rats	inflammatory cytokines inhibition through the blockade of Sec61 translocon (see also [61])	Purified strain 1615	[63]


↓: decrease; ↑: increase; K+: potassium; AT2R: angiotensin II receptor 2; ∑: synthetic; DRG: dorsal root ganglion; MycoL: mycolactone; Ref: reference; LPS: lipopolysaccharide.

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
