# Peer review of "Could Mycolactone Inspire New Potent Analgesics? Perspectives and Pitfalls"

_toxins, 2019, doi:10.3390/toxins11090516_

Round 1

Reviewer 1 Report

Very interesting review on the potential use of mycolactone as a novel painkiller. With regards to the synthesis of mycolactone, the complexity of its biosynthesis is clear, but would it be possible to create a bacterial recombinant strain (e.g. of E. coli) expressing the genes required to produce mycolactone? Alternatively, would it be possible to re-engineer the fast growing Mycobacterium smegmatis to produce this toxin? Presumably, some of the required genes may be conserved in M. smegmatis or other fast-growing mycobacteria. After all, mycolactone-like metabolites are produced by other Mycobacterium spp. (Line 300). I think that this should be discussed in the text as a future indication. Otherwise, I have only some minor comments with regards to the format or some parts of the text. Please, check that all species names are written in italics, and something happened with the text starting in line 280. Finally, the footnote of Figure 2 is misplaced.

Author Response

We would like to warmly thank the Reviewer#1 for pointing out the significance of our work and for his critical advices to improve our manuscript, which we took into full account.

We thank the Reviewer#1 for raising this point on the engineering of mycolactone in heterologous systems. Indeed, this has been previously investigated by the group of Timothy Stinear. The text has now amended as follows “A recurrent idea, since the discovery of the giant plasmid pMUM001 by Stinear and Brosch, harboured by M. ulcerans, that encodes mycolactone [104], was to express the plasmid in the faster growing and closely related mycobacterium Mycobacterium marinum. However, the genetic background of M. ulcerans appeared to be important for the production of mycolactone as despite efficient production of mycolactone polyketide synthases (PKS), whole mycolactone could not be detected properly expressed in M. marinum [105]. In line with this, the same group led by Stinear reported that fast growing Mycobacterium smegmatis, which is a widely used host for mycobacterial product expression was unsuitable for efficient replication of PKS encoding plasmids required for mycolactone synthesis [105]. To our knowledge, it has not been reported as yet any successful expression of mycolactone in Escherichia coli.”

Next, we agree with the reviewer that mycolactone-like metabolites production by other mycobacterium ssp would need further investigations, and that these metabolites could be tested to examine whether they display the same mode of action of mycolactone and if they exhibit analgesic properties. Appropriate credit to this point is now given in the manuscript and the text was modified accordingly. The text now reads “Also, mycolactone-like metabolites are produced by other mycobacteria fish and frog pathogens, referred to as “mycolactone-producing mycobacteria” such as M. liflandii, M. pseudoshottsii, or M. shinshuense [111-113]. These strains, considered as M. ulcerans ecovars, could be considered for mycolactone production, but mycolactones produced by these strains appear to be structural variants of the mycolactone A/B produced by M. ulcerans [114]. M. liflandii, M. pseudoshottsii, or M. shinshuense thus produce mycolactone E, F, S1 and S2 [111-113], and it would be necessary to further assess these different mycolactones for their analgesics properties.”

We paid attention to correct species names when not written in italics; we edited the text beginning from line 280 to 296 which was in a wrong font size and positioned the footnote of Figure 2 at the right place. We thank the reviewer for pointing out these mistakes.

Reviewer 2 Report

(1) The English of this manuscript must be improved before resubmission

(2) Authors listed different kinds of small molecules (e.g., EMA401, C21, capsaicin, etc.) or peptides for their potential therapeutics of pain, I would suggest authors add another figure to show their structures, bioactivity, efficacy, etc.

(3) The font size from line 280 to 296 seemed different, please fix it

(4) Target(s) or off-targets of mycolactone may help to comprehensively understand its role, I would suggest authors add a discussion

(5) The angiotensin pathway has been identified as another target for mycolactones. However, the morphological and functional effects of mycolactone are not affected by angiotensin II or AT2R antagonist EMA401. How do the authors explain this?

Author Response

We thank the Reviewer#2 for the evaluation of our work. We have carefully taken into account the suggestions for improvement for the revision of our manuscript.

The English of this manuscript must be improved before resubmission. We carefully edited the manuscript. The changes are now highlighted in blue in the manuscript. Authors listed different kinds of small molecules (e.g., EMA401, C21, capsaicin, etc.) or peptides for their potential therapeutics of pain, I would suggest authors add another figure to show their structures, bioactivity, efficacy, etc. We thank the reviewer for this suggestion, and accordingly we added a table (referred to as table 2) to summarize the structure and effects of the main molecules mentioned in the manuscript. The font size from line 280 to 296 seemed different, please fix it. The font size was corrected for this paragraph; we thank the reviewer for pointing out this problem. Target(s) or off-targets of mycolactone may help to comprehensively understand its role, I would suggest authors add a discussion. We agree with the reviewer that an effort to clearly emphasize the targets of mycolactone, that impact on its physiological effect would indeed help understanding the role of the toxin. A paragraph at the beginning of the section 2 “Mycolactone-induced analgesia: inflammatory versus non inflammatory context” was added accordingly: From Line 141, the text now reads: “Several cellular targets were proposed to account for the various effects of mycolactone. The first one was the Wiskott-Aldrich Syndrome protein (WASP) reported by the Demangel group [52,59,60]. Indeed, this group showed that mycolactone binds to WASP, with an affinity 100-fold greater than its natural activator, leading to alterations in actin dynamics, defective cell adhesion and eventually apoptosis [60]. Mycolactone-driven destruction of cutaneous tissues was thus suggested to be caused by hyper-activation of neural (N)-WASP. The second target that was identified by Simmonds et al. is the core subunit of the Sec61 complex, being now recognized as the primary cellular target of mycolactone [54-56, 59]. As Sec61 is responsible for the transport of newly synthesized proteins through the endoplasmic reticulum membrane, the toxin was shown to impact on the expression of a set of proteins such as immune mediators. In parallel, the groups of Altmann and Pluschke showed that mycolactone acts as an inhibitor of the signalling of the Target of Rapamycin (mTOR), preventing Bim-dependent apoptosis [61]. For instance, mycolactone was shown to inhibit the mTOR-dependent phosphorylation of the serine/threonine protein kinase Akt, leading to the up-regulation of transcription factor Fox03 and the increase of Bim expression. The increase of Bim expression leads to cell apoptosis. This signalling was shown to be important in BU pathogenesis with Bim knockout mice not developing necrotic lesions of BU when infected with M. ulcerans [61]. Simultaneously, the groups of Brodin and Marsollier identified AT2R as being the main receptor involved in mycolactone-mediated analgesia [52]. More recently, a modelisation study predicted that mycolactone would bind to Munc18b, a key chaperone protein of blood platelets degranulation, blocking exocytosis and hence impairing the release of compounds and factors that are critical for wound healing processes [62]. In the present review, we describe the mechanisms proposed for sustaining mycolactone-induced analgesia.” The angiotensin pathway has been identified as another target for mycolactones. However, the morphological and functional effects of mycolactone are not affected by angiotensin II or AT2R antagonist EMA401. How do the authors explain this? The signalling pathways involving AT2R are still poorly characterized and no standard functional assays to study the pharmacology of AT2R ligands. Thus we do not understand yet why mycolactone effects are not affected by angiotensin II or AT2R antagonist EMA401. We hypothesize that the binding site of mycolactone differs from the one of other ligands of AT2 receptor such as angiotensin II or EMA401, and that the pathway triggered by mycolactone is specific of the molecule and not shared by others ligands. Further investigations would be needed to dissect in detail mycolactone binding to the AT2R receptor. The text, L204 is now: “AT2R is an unusual GPCR, with the lack of observed G proteins or recruitment of β-arrestins and finally absence of intracellular internalization. Accordingly, the first report of an AT2R structure bound to a ligand showed that helix VIII was found in a non-canonical position stabilizing the active-like state and thus preventing coupling to G-proteins [73]. It is thus difficult at present to undertake mechanistic studies to compare the pharmacology of the different known ligands of AT2R. Danser and Anand showed that AT2R signalling cascade in sensory neurons elicits peripheral pain sensitization and that the AT2R ligand EMA401 alleviated pain in preclinical models [74,75]. Pain relief was further demonstrated in a phase II clinical trial involving patients with neuropathic pain associated with postherpetic neuralgia [33,76]. As a difference to EMA401 or other known ligands of AT2R (including the endogenous peptide AngII or the C21 agonist) only mycolactone appeared able to trigger hyperpolarization of DRG neurons [32,66,77,78]. Moreover, EMA401 or C21 did not inhibit hyperpolarization triggered by mycolactone [66] (Table 2). This observation pinpoints the specificity of mycolactone/AT2R interactions, supporting the hypothesis that the AT2R binding site of mycolactone is different from that of AngII, C21 or EMA401.”

Round 2

Reviewer 2 Report

All my concerns have been addressed. I recommend publishing this manuscript in the present form